# A Retrospective Analysis of Admission Trends and Outcomes in a Wildlife Rescue and Rehabilitation Center in Costa Rica

**DOI:** 10.3390/ani14010051

**Published:** 2023-12-22

**Authors:** Maria Miguel Costa, Nazaré Pinto da Cunha, Isabel Hagnauer, Marta Venegas

**Affiliations:** 1Veterinary Medicine Faculty, Lusófona University, 1749-024 Lisbon, Portugal; 2Rescate Wildlife Rescue Center, Fundación Restauración de la Naturaleza, Apdo, Alajuela 1327-4050, Costa Rica

**Keywords:** wildlife, release, mortality

## Abstract

**Simple Summary:**

We reviewed the database of a wildlife rescue and rehabilitation center in Costa Rica and described the main causes of admission, the admission factors that influenced release and mortality, and the predictive factors of survival and mortality of wildlife to determine general patterns and relevant factors currently affecting wildlife in Costa Rica. The results of the study demonstrate the value of maintaining, improving, and studying databases from wildlife rehabilitation centers to lead to a better understanding of threats to wildlife and subsequent implementation of conservation actions.

**Abstract:**

The evaluation of data regarding rehabilitation practices provides reference values for comparison purposes among different rehabilitation centers to critically review protocols and efficiently improve each center. The aim of the present work was to present the main causes of admission to Rescate Wildlife Rescue Center for each taxonomic group, to determine the admission factors that influenced the release and mortality, and to determine the predictive factors of release and mortality of wildlife. To this end, a retrospective study was carried out based on 5785 admissions registered in the database of Rescate Wildlife Rescue Center in Costa Rica in 2020 and 2021. Statistical analysis consisted of sample characterization via the analysis of several categorical variables: species, order, class, age group, cause of admission, outcome, clinical classification and days in the hospital, and respective association with the mortality or release rate. Most of the rescue animals were birds (59.3%), then mammals (20.7%), reptiles (17.4%), and finally ‘others’ (2.6%). The main causes of admission were ‘captivity’ (34.9%), ‘found’ (23.3%), and ‘trauma’ (19.3%). Animals rescued due to ‘captivity’ and the classes ‘birds’ and ‘reptiles’ had the highest release rates. The causes of admission ‘trauma’ and ‘orphanhood’ and the class ‘birds’ had the highest mortality rates. In general, a greater number of days spent in the hospital and membership in the classes ‘reptiles’, ‘juveniles’, in need of ‘basic care’, or ‘clinically healthy’ were predictors of survival. In contrast, the age groups ‘infant’ and ‘nestling’ were predictors of mortality. These results demonstrate the value of maintaining, improving, and studying databases from wildlife rehabilitation centers, as they can provide useful information that can be used to enhance the allocation of economic resources, treatment methods, disease surveillance, public education, and regulatory decision-making, leading to a better understanding of threats to wildlife and subsequent implementation of conservation actions.

## 1. Introduction

The International Wildlife Rehabilitation Council (IWRC) defines wildlife rehabilitation as ‘the temporary care of injured, diseased, and displaced indigenous animals and the subsequent release of healthy animals to appropriate habitats in the wild’ [1]. All animals must be released into the wild with a chance of survival equal to other free-living members of the species. Their survival depends on the animal being healthy and having an appropriate behavior to adapt to living in the wild. The alternatives are permanent captivity or euthanasia (euthanasia should be considered at all stages of assessment and treatment, to avoid unnecessary suffering) [2].

Conservation medicine was developed in response to the emergence of new diseases and threats to human and animal health arising from ecological and anthropogenic changes. It examines ecology-related health problems, including emerging diseases; the biological effects of pollutants; and the implications that ecological changes such as fragmentation, simplification and degradation of habitats, loss of biodiversity, and global climate change have for health [3]. Wildlife rehabilitation centers should contribute to ecosystem health monitoring as they are in a unique position to observe ecological changes and anthropogenic effects on wildlife health [3].

To assess the success of rehabilitation, we first must define success, which is viewed differently by various people involved in the process. Most people measure the process of success by the number of cases rehabilitated, along with the educational benefits of the process [4]. In addition, several aspects must be considered critical for the success of the process, including the physical development and the acquisition of a natural behavior of wild animals [5]. The most important factor across all species in predicting whether a wild animal will eventually be released is the severity of the injury or illness at admission. The more severe the injury or disease, the less likely animal is to be released, let alone survive in the wild [6,7]. There are many considerations to make when caring for wildlife. Being under care for any period can induce stress in wild animals, caused by proximity to humans and new stimuli in captivity, which can affect the chances of survival [6]. Chronic stress can also have harmful immunological consequences, which can also affect the recovery process [6,8], so it is important to follow protocols. Standardization of protocols has been established to ensure animal welfare during rehabilitation [9] and to increase the likelihood of animals being released after rehabilitation, as well as to increase the success of their life in the wild after rehabilitation [1,10]. 

The analysis of data on rehabilitation practices is essential to provide reference values with the aim of comparing rehabilitation centers to critically analyze protocols and efficiently improve, if necessary, each center [5]. There are several studies that have performed a retrospective analysis of data derived from rehabilitation centers to critically assess the rehabilitation process [5,11,12]. Data analysis can provide useful information on factors such as taxonomic group, age group, and causes of admission and their effect on successful rehabilitation of different species [12]. Results can generate evidence-based recommendations to minimize animal suffering, improve rehabilitation success as well as the survival and reproductive success of released individuals (particularly in the case of species of conservation interest), effectively invest limited resources and to assess the health of free-ranging populations [5,11,12]. There are currently thousands of rehabilitation centers around the world; they have an abundant amount of raw data, which can be used by researchers, but access to these data is limited, and data comparison between centers is complicated due to the lack of standardization [5]. 

Animals’ individual welfare must be the priority, and this requires careful handling. Treatment of rescue wildlife must be kept to the absolute minimum if they have any hope of returning to the wild [2]. The longer an animal is dependent on its keepers, the more attention needs to be given to proper preparation for release and to ensure, for example, that it is able to successfully forage or hunt in the wild. Furthermore, for social species, prolonged time in captivity can have deleterious effects on acceptance by conspecifics after release [2]. 

Here, we describe the general characteristics of wildlife rescues for birds, mammals, and reptiles in our study area, using both descriptive statistics and statistical models. The objectives of this study are the following: (1) to determine the main causes of admission to the Rescate Wildlife Rescue Center (RWRC) in Costa Rica for each taxonomic group (birds, mammals, and reptiles); (2) to determine their admission factors (class, clinical classification, age group, and days in hospital) that influence release; (3) to determine the admission factors that influence mortality (euthanasia and death under/prior to care) of the center animals; and (4) to determine which factors predicted the likelihood of mortality.

## 2. Materials and Methods

### 2.1. Study Area

A retrospective study of wildlife rehabilitation was performed using data from the Rescate Wildlife Rescue Center (RWRC) in Alajuela, Costa Rica. The center receives animals from Costa Rica, mainly from the Alajuela region. Costa Rica is a country located in Central America and has a tropical climate. In addition, it has a permanent population of around 5 million people and an area of 51,100 km^2^.

The work described in this study did not require approval from an Animal Ethics Committee, as it deals only with data.

### 2.2. Data Preparation

The wildlife rehabilitation providers collected data about each rescued animal using Microsoft Office Excel 2010^®^ (Microsoft, Redmond, WA, USA) spreadsheets. For each animal, the following information was noted: common name, taxonomic group (species), and age group; also included was information about the rescue (date, location, region, cause of admission, type of injury), days in the hospital, outcome, and place of release.

All database records were reviewed for accuracy and formatting consistency. Simple data entry errors by the wildlife rehabilitators (such as minor incorrect spelling of species or place names) were corrected. Subsequently, for this study we used data from two consecutive years (from 1 January 2020 to 31 December 2021). The statistical analysis consisted of characterizing the sample via the analysis of the various categorical variables: taxonomic group (species, order, and class), age group, clinical classification, cause of admission, outcome, and days in the hospital. 

It should be noted that the data for the variables age group, clinical classification, and days in the hospital were missing for the year 2020. Consequently, these variables are recorded as missing data in the graphs and/or tables that include these variables. 

### 2.3. Animal Classification

For analytical purposes, reported species were grouped in the following broader taxonomic categories (classes and orders): (1) birds (Anseriformes, Apodiformes, Caprimulgiformes, Charadriiformes, Ciconiiformes, Columbiformes, Coraciiformes, Cuculiformes, Falconiformes, Passeriformes, Pelecaniformes, Piciformes, Psittaciformes, Strigiformes, Trogoniformes); (2) mammals (Artiodactyla, Carnivora, Chiroptera, Cingulata, Didelphimorphia, Lagomorpha, Perissodactyla, Pilosa, Primates, Rodentia); (3) reptiles (Crocodylia, Lacertilia, Serpentes, Testudines), and (4) others, including amphibians (Anura), arthropods (Lepidoptera, Decapoda), fishes, and unknown. 

The age for birds was categorized as the following: nestling, fledgling, juvenile, adult and unknown. For mammals and reptiles, the categories for age were the following: neonate, infant, juvenile, adult, and unknown. 

### 2.4. Cause of Admission Classification

The reported causes of admission were grouped into the following main categories: (1) trauma (associated with an anthropogenic activity, such as collision with vehicles, buildings or other human structures, power lines, fences, and electrocution); (2) captivity (animals maintained in captivity and/or confiscated by rangers or the police due to poaching or the illegal pet trade); (3) found (animals accidentally found in wrong places, such as buildings or other human-made structures or orphaned due to unnecessary human intervention); (4) orphanhood (animals that need their parents to survive who were abandoned by their parents or whose parents were killed); (5) environmental pollutants (crude oil, poison, or toxin ingestion); (6) disease (infectious, parasitic, metabolic, or nutritional); (7) transferred (transferred from another licensed wildlife rehabilitator in Costa Rica); and (8) unknown.

### 2.5. Outcome Classification

The variable outcomes included: (1) died under/prior to care; (2) euthanized; (3) released; (4) permanent care (placed in the RWRC sanctuary); (5) under care (still under care or transferred to another licensed wildlife rehabilitator in Costa Rica); (6) escaped; and (7) unknown. 

A binary score of animal fate (‘mortality’ or ‘survival’) was created to model the likelihood of mortality. ‘Mortality’ includes animals that died prior to/under care or were euthanized. ‘Survival’ includes animals who were released, placed in permanent care, remained under care, escaped, or had an unknown outcome. 

### 2.6. Clinical Classification

The clinical classification variable was based on the primary diagnoses at the moment of the admission, grouped into the following categories: (1) basic care (in need of care to survive); (2) clinically healthy; (3) physical injury (injury caused by trauma from an external force); (4) clipped feathers; (5) dermatological disease; (6) neurological disease; (7) nutritional disease; (8) ocular disease; (9) infectious disease; (10) gastrointestinal disease; (11) urogenital disease; (12) respiratory disease; (13) tumors; (14) petrochemical exposure; (15) intoxication; (16) nonspecific disease (not assignable to a particular category or classification); and (17) unknown;

### 2.7. Statistical Analyses

Regarding the interferential statistical analysis, the SPSS v25.0 (IBM Corporation, Armonk, NY, USA) program was used. Descriptive statistics are presented as N (%) for discrete variables and mean and standard deviation for continuous variables. Statistical inference used the unpaired *t*-test or ANOVA tests for continuous variables, depending on whether two or more groups were being compared, respectively, and the χ^2^ test to compare two or more groups in the case of discrete variables. Whenever appropriate, post hoc Sidak or Bonferroni correction for multiplicity was used. A multivariate logistic regression was also carried out to determine mortality. In all cases, *p* < 0.05 was considered the significance threshold. 

## 3. Results

### 3.1. Taxonomic Groups

Over the two-year (2020–2021) survey period, a total of 5785 reports were submitted by wildlife rehabilitation providers. In the year 2020, 2270 animals were rescued, and in the year 2021, 3015 animals were rescued, demonstrating a slight increase (8.8%) in the number of rescues. Also, in 2021, more birds (*p* < 0.001) and mammals (*p* < 0.001) were rescued than in 2020, but fewer reptiles (*p* < 0.001) and others (*p* < 0.001). The increase can be attributed to bird and mammal rescues (Figure 1). The rescues were primarily birds (59.3%), followed by mammals (20.7%), reptiles (17.4%), and lastly others (2.6%), which includes amphibians, arthropods, and unknown species (Table 1). In total, 315 species were reported across all classes. Birds were the most species-rich category (187 species), followed by reptiles (60 species), mammals (41 species), and others (27 species) (Table 1). Within the category of birds, the most affected orders were Psittaciformes (20.4%) and Passeriformes (18.7%). For mammals, the most affected order was Didelphimorphia (10%). Finally, in reptiles the order Testudines (11.9%) was the most affected (Table 2). 

### 3.2. Causes of Admission

Overall, three causes of admission were dominant across all taxonomic groups: ‘captivity’ (34.9%), ‘found’ (23.3%), and ‘trauma’ (19.3%) (Table 3). Within each class, there were several significant differences between the number of animals rescued per cause of admission. In mammals, the two main causes of admission were ‘orphanhood’ (*n* = 422; *p* < 0.001) and ‘found’ (*n* = 377; *p* < 0.001). Moreover, ‘captivity’ was the main cause of admission in birds (*n* = 1450; *p* < 0.001), as well in reptiles (*n* = 494; *p* < 0.001). As for the group ‘others’, the main cause of admission was ‘transferred’ (*n* = 145; *p* < 0.001). It is worth mentioning that we only considered causes of admission applying to over 10% of rescued animals (Figure 2).

### 3.3. Clinical Classification

The clinical classification data was missing in 47.9% of the rescue animals. In the remaining 52.1%, it should be noted that 17.0% of the animals were considered ‘clinically healthy’, 16.2% needed ‘basic care’, and 10.5% had a ‘physical injury’ (Table 4).

### 3.4. Outcome Analyses

Regarding the outcomes of the rescue animals, 41.8% (*n* = 2419) were ‘released’, 24.1% (*n* = 1394) ‘died under/prior to care’, 23.7% (*n* = 1372) were ‘euthanized’, and 7.5% (*n* = 432) went to ‘permanent care’. It should be noted that only 4 animals had an unknown outcome and only 12 animals ‘escaped’ the center. 

When the outcome rates were stratified by cause of admission and class, animals with an unknown outcome were excluded. Within the category of mammals, the most notable outcome was ‘euthanized’ (*n* = 741; *p* < 0.001). However, in the categories of birds (*n* = 1420; *p* < 0.001), reptiles (*n* = 728; *p* < 0.001), and others (*n* = 89; *p* < 0.001), the outcome ‘released’ was the most frequent outcome in these classes, having a significantly superior rate to all other outcomes for birds and reptiles (Figure 3). 

The average of number of days in the hospital was higher for the outcome ‘under care’ (66.9 days; *n* = 14) followed by ‘permanent care’ (56.1 days; *n* = 102), ‘released’ (47.3 days; *n* = 1104; *p* < 0.001), ‘escaped’ (33.3 days; *n* = 4), ‘died under/prior care’ (14.4 days; *n* = 542; *p* < 0.001), and finally ‘euthanized’ (2.5 days; *n* = 787; *p* < 0.001), which was significantly lower when compared to all the others. There was a wide variability of days in the hospital for all outcomes (Appendix A).

When the outcomes rates were stratified by cause of admission, the ‘trauma’ (41.6%; *p* < 0.001) category showed the highest ‘euthanized’ rate. Meanwhile, the ‘orphanhood’ (4.4%; *p* < 0.001) category had the lowest rate of unassisted death. On the other hand, the cause of admission that showed a significant higher release rate was ‘captivity’ (52.0%; *p* < 0.001). Furthermore, animals that went to ‘permanent care’ were also primary admitted due to ‘captivity’ (73.9%; *p* < 0.001) (Figure 4). 

### 3.5. Fate and Likelihood of Release

Looking into more detail at the outcome ‘released’, it was dependent upon the taxonomic group (χ^2^ (5) = 796.27, *p* = 0.000). The proportion of rescues that were released to that of those that were not released was substantially higher for birds (58.7%) than for reptiles (30.1%), mammals (7.5%), and others (3.7%) (Table 5). 

The outcome ‘released’ was also dependent on the cause of admission (χ^2^ (6) = 949.07, *p* = 0.000). Animals rescued due to ‘captivity’ (61.2%) or ‘transferred’ (58.3%) showed the highest percentages of release, and those rescued due to ‘trauma’ had the lowest percentage of release (12.9%). It is noteworthy that we only considered the causes that applied to more than 10% of the total (Table 6). 

An association between release and clinical classification was observed (χ^2^ (15) = 793.30, *p* = 0.000). Rescue animals that were ‘clinically healthy’ (71.9%) showed the highest release rate, followed by those in need of ‘basic care’ (25.7%), ‘nonspecific disease’ (18.5%) and finally by those admitted due to ‘physical injury’ (10.9%); only variables above 5% were considered (Table 7). 

Rescue animals in the ‘adult’ (34.8%), ‘juvenile’ (41.4%), and ‘fledgling’ (37.9%) age groups had higher release rates than animals in the ‘infant’ (15.0%) and ‘nestling’ (20.8%) age groups. an association between release and age group It was observed (χ^2^ (6) = 106.79, *p* = 0.000) (Table 8). 

Finally, it is worthy of mention that in 2021, the average number of days in the hospital was significantly lower for non-released animals (11.5 days; *p* < 0.001) when compared to released animals (47.2 days; *p* < 0.001) (Appendix A). 

### 3.6. Fate and Likelihood of Mortality

The ‘died under/prior to care’ and ‘euthanized’ disposition categories were combined to model the likelihood of mortality. Mortality was dependent upon class (χ^2^ (5) = 1070.70, *p* = 0.000); birds had the highest percentage of mortality (59.4%), followed by mammals (34.3%), reptiles (4.1%), and others (2.1%) (Table 9). 

An association between mortality and cause of admission was observed (χ^2^ (6) = 1754.14, *p* = 0.000). Animals rescued due to ‘trauma’ (83.9%), or ‘orphanhood’ (83.7%) had the highest percentages of mortality, followed by ‘found’ (54.4%), and finally, ‘captivity’ (17.3%) had the lowest percentage of mortality. It should be noted that we only considered causes that applied to more than 10% of the total animals (Table 10).

When regarding clinical classification, we observed an association with mortality (χ^2^ (15) = 1202.03, *p* = 0.000). Animals that were ‘clinically healthy’ (9.3%) had the lowest mortality rate when compared to all other clinical classifications; only variables above 5% of the total were considered (Table 11). 

In addition, mortality was also dependent on the age group (χ^2^ (6) = 142.42, *p* = 0.000). Animals in the ‘adult’ (52.5%), ‘juvenile’ (50.8%), and ‘fledgling’ (58.4%) age groups had lower mortality rates than animals in the ‘infant’ (78.6%) and ‘nestling’ (77.9%) age groups (Table 12). 

Finally, in 2021, the average number of days in the hospital was significantly lower for animals that died under/prior to care or were euthanized (7.2 days; *p* < 0.001) compared to animals that survived (48.2 days; *p* < 0.001) (Appendix A). 

### 3.7. Predicted Likelihood of Mortality vs. Survival

According to the parameters of the multivariate logistic regression that was used to evaluate the predictive factors of mortality, the greater the number of days spent in the hospital, the greater the probability of survival. Additionally, membership in the categories ‘reptiles’, ‘juveniles’ ‘basic care’, or ‘clinically healthy’ were also predictors of survival, as expB < 1 is favorable to death. In contrast, membership in ‘infants’, or ‘nestlings’ were predictors of mortality (Table 13).

## 4. Discussion

Over the two-year period (2020–2021), birds were the most admitted native animals at the center, followed in smaller proportions by mammals, and, lastly, reptiles. The main causes of admissions to the center were ‘captivity’, ‘found’, and ‘trauma’. 

In birds, the order Psittaciformes was the most affected, and the main cause of admission was captivity (e.g., as pets). According to Drews (2000), 23.5% of the houses in Costa Rica had wild animals and the majority were illegally kept, with parrots being the main species [13]. In both urban and rural areas, keeping a wild animal is a custom that has been passed down from generation to generation. Whatever the reason that has encouraged the ownership of wild animals, captivity is causing impacts on natural populations, which, added to other factors, puts animals on the verge of extinction (for example, wild cats and parrots) [14]. 

In mammals, the order Didelphimorphia was the most affected (10% of the admitted animals) due to the large number of common opossums (*Didelphis marsupialis*) that are admitted to the RWRC. The main cause of admission in mammals was orphanhood and being found. The common opossum is still considered a harmful and undesirable animal in some peri-urban areas, causing human–wildlife conflicts [15]. This species does not present an imminent risk of extinction, as its population size is estimated to be large and it has a high tolerance to habitat transformation, mainly reflected in its ability to use alternative resources in urban environments [16]. Thus, we can say that people consider opossums harmful and a pest. Consequently, when people find these animals on their properties, they hand them over to the RWRC, leading to increased admission of ‘found’. People also end up killing adult possums, leading to an increased admission of orphans to the RWRC. In addition, well-meaning members of the public often find young wild animals that have fallen from trees or appeared mysteriously and assume they are orphans. It is necessary to determine whether the animal is truly an orphan, as the animals found would have a better chance of surviving in the wild with their parents than in the rehabilitation center. Most animals found end up being euthanized due to lack of resources or die during or before treatment [17]. Enhancing public outreach efforts to raise awareness about the causes of wildlife injury and promoting understanding of natural behaviors could play a pivotal role in mitigating these numbers.

In reptiles, the main cause of admission was also ‘captivity’, the order Testudines being the most affected. Most wild animals rescued in the neotropics are the product of apprehension of live animals intended mainly for the pet trade. Birds (mainly parrots) and reptiles predominate among the rescued fauna. Among reptiles, seizures are predominantly of iguanas and turtles destined mainly for the pet market and in some cases for local consumption [18]. Some of the major medical problems that are anticipated for neotropical fauna are due to the stress and malnutrition associated with captivity under extreme conditions (for example, seized shipments of hundreds of animals crammed together) [18].

The analysis of the outcomes showed a general release rate of 41.8% of admissions. This rate is similar to those of other studies, such as the release rate reported by the RSPCA in England, which was around 40% [7], and a study conducted on rehabilitation centers in Australia, where overall release rates ranged from 38% to 45% [19]. Evaluation of release rates by cause of distress and species provides useful information to rehabilitators or veterinarians to determine the likelihood of successful rehabilitation for an individual case or to allocate resources [20].

Birds were the class with the highest release rate, followed by reptiles. Also, the main cause of admission of birds and reptiles was ‘captivity’. The analysis of the outcomes showed that ‘captivity’ had the highest release rate. Rescued animals admitted to the center due to ‘captivity’ are mostly healthy adults or juveniles that do not need treatment, as is confirmed by the large percentage of released of clinically healthy animals. According to a study on wildlife rehabilitation centers in Catalonia, a high percentage of released animals rescued from captivity is attributed to the high percentage of healthy animals [5]. In fact, the severity of the clinical condition has been reported across all species as the most important predictive factor for determining whether a wildlife casualty will survive to be released from a rehabilitation center [6,20,21].

The analysis of rehabilitation outcomes showed that ‘trauma’ and ‘orphanhood’ had the highest percentages of mortality. Trauma related to anthropogenic activities represented the most frequent cause of mortality, consistent with previously published literature [20,21,22,23]. Furthermore, ‘traumatic injury’ was the clinical classification most associated with mortality. The most common causes of trauma observer in the RWRC were attack by domestic animals and collision with man-made structures. This demonstrates the impact of anthropogenic factors; therefore, improving public outreach regarding causes of wildlife injury and natural behaviors may help to reduce these numbers. For example, by teaching people that free-roaming domestic animals can cause an impact on wildlife populations, we may influence people to protect wildlife. 

The second cause of admission with the highest mortality was ‘orphanhood’. Of all the animals admitted due to this cause, the majority were ‘euthanized’, highlighting the euthanasia of mammals. Orphans are animals that use the most economic and human resources, as they need full-time basic care. In the RWRC, due to lack of resources, animals that needed basic care were selected for their value in the conservation of wild populations, with endangered wild animals having more value, and were also selected for their representative value in society. Thus, the ‘orphaned’ item represents a potential bias, since part of the mortality outcome was determined by a decision other than the actual recovery potential. 

The second clinical classification most associated with mortality was ‘non-specific disease’, which can be explained by the center’s lack of economic resources to arrive at a specific diagnosis. Therefore, due to lack of funding, it is not always possible to determine the cause of death from infectious or parasitic diseases or chronic poisoning, which are potentially included in the non-specific disease group. Thus, in this group we will have a set of diseases that potentially determine a bias in the study. 

We can also see that the need for ‘basic care’ is the third clinical classification most associated with mortality. Basic care applies to animals that still need their parents to survive. The age groups that need basic care are infants, nestlings, fledglings, and juveniles. Caring for these animals is a challenging task due to the heterogeneity of species and diets and the inherent fragility of pediatric patients [24]. Furthermore, humans are very inadequate surrogate parents for wild young, despite their adequate ability to feed them. There is much more investment by parents in their offspring than just food. Species recognition, sibling interaction and rivalry, and learning wild food sources are just a few of the critical skills needed for successful survival [23]. 

Released animals spent an average of 47.2 days at the rehabilitation center, while non-released animals spent a significantly lower average at the center. In agreement with Molony et al. (2006), this suggests that temporary captivity improves the chances of survival, as it allows for the accumulation of fat reserves and reduces the stress suffered during release manipulation [25]. However, most studies state that longer periods of rehabilitation lead to the loss of wild behaviors such as avoiding predators and the interruption of social development due to human habituation, which can result in low survival after release [10,26,27,28,29]. 

Animals that survived spent significantly more days in the hospital, an average of 48.2 days, when compared with the animals that died or were euthanized. Animals that died spent less time in the clinic, since triage decisions such as euthanasia must be made quickly, ideally within 48 h of admission, to avoid unnecessary suffering [30]. Although authors have stated that long periods in captivity had a negative effect on the chances of survival and release of animals, due to the negative impact that captivity can have on stress levels and, subsequently, on immune function and health [31,32]. In this study, we concluded that the greater the number of days in the hospital, the greater the probability of survival. Perhaps, animals that survived to be released generally spend more time in captivity receiving treatment than animals for which treatment is unsuccessful and have died or been euthanized. 

While longer periods in care allow animals to recover successfully, prolonged captivity can negatively affect their survival after release, for example, by decreasing their wariness of predators and dangers [33]. Therefore, the impact of captivity on the physiological stress levels and recovery time of rescue wildlife is an area that deserves further research. Being under care for any period can induce stress in wild animals, caused by proximity to humans and new stimuli in captivity, which can affect the chances of survival [6]. Chronic stress can also have harmful immunological consequences, which can also affect the recovery process [6,8]. As welfare is one of the RWRC’s priorities, handling is kept to a minimum to avoid animals developing dependence on their keepers and chronic stress, so a good triage is of most importance to evaluate the need time of treatment. Therefore, it is important to follow protocols. Standardization of protocols has been established to ensure animal welfare during rehabilitation [9] and to increase the likelihood of animals being released after rehabilitation, as well as to increase the success of their life in the wild after rehabilitation [1,10].

In this study, animals from the adult, juvenile, and fledgling age groups had the highest release rates. The age group on arrival will influence the release rate of animals from the center. However, it depends on the clinical condition that the animals had on arrival at the center [12]. Animals in the infant and nestling age groups had the highest mortality rates. As mentioned above, the younger the animals, the more fragile the animals, leading to an increase in the outcome ‘died’. Simpson and King (2017) state that juvenile animals can be more difficult to rehabilitate than adults, and therefore rehabilitation may not be feasible, but our analysis found that juveniles are more likely to survive than other age groups [34]. This could be explained by the fact that, because most of the rescue animals are nestling or juvenile Psittaciformes, they are grouped with animals from the same species, and consequently they learn from each other and can form better bonds with conspecifics.

The main cause of admission of animals in the RWRC with the outcome ‘permanent care’ was captivity. Other reasons for permanent care were animals of non-native species that were confiscated or voluntarily surrendered. In addition, the permanent care of birds stood out. Most animals admitted due to captivity were born in captivity and lacked the skills to survive in the wild. They may be habituated to humans (which can be deadly for them), and many animals admitted are older or suffer from health problems. Thus, their reintroduction in natural habitats is impractical [35]. Additionally, these animals may then be raised in inappropriate conditions, increasing to the possibility of zoonotic disease transmission [19]. Occasionally it is justifiable to keep wild animals that cannot be released in sanctuaries or educational settings [1], which include captive breeding programs of rare or endangered species, for educational purposes, or for use as imprinting models for young animals of the same species to allow for breeding without imprinting on humans [2]. 

Our study has shown that clinical classification is one of the predictors of survival; that is, needing basic care or being clinically healthy were predictors of survival. Agreeing with Molony et al. (2007), this says the most important predictive factor for determining whether a rescued wild animal survives to be released from a rehabilitation center, across all species, is the severity of injury or illness symptoms: the more severe the injury or illness, the less the probability that the individual will survive [6]. Thus, it is of great importance to carry out triage based on clinical diagnosis when animals are admitted. More resources should be dedicated to animals that will benefit the most, thus reducing the potential for prolonged suffering in animals that are less likely to survive until release and improving the welfare of animals in rehabilitation and the survival rate after release [6]. 

Molina-López et al. (2017) also categorized the severity of injuries, with injuries rated ‘very severe’ (major injuries, emaciation, paralysis, blindness, respiratory distress) being related to lower release rates [5]. The possibility of categorizing diseases/injuries according to their severity in the RWRC database would add value to examining their influence on outcomes in future studies.

The establishment of standardized protocols for rehabilitation reports and the systematic collection of digital data would significantly enhance the efficiency of compilation and analysis. This approach is vital in preventing the loss of valuable information and, more importantly, in fostering long-term trend studies. Such studies are crucial for gaining a deeper understanding of the multifaceted impacts of human activities and climate change on rescued wildlife. Additionally, advocating for the adoption of international guidelines is paramount, since more data from standardized records on the activity of wild animals means more information for the future, to improve the chances of success for the animals. 

The utilization of a comprehensive database enables rescue centers not only to educate the public but also to provide compelling evidence that anthropogenic actions significantly impact wildlife populations, potentially reducing the number of admitted animals. Additionally, it serves as a crucial tool for advocating the allocation of financial resources needed to diagnose diseases, including zoonotic ones, thereby mitigating the spread of infections. In terms of resource allocation, the database aids in optimizing the efficiency of rescue centers by streamlining the triage process, facilitating decisions regarding which animals would benefit from treatment and those that may be more compassionately euthanized, thus preventing unnecessary suffering.

Several limitations to the current study are recognized. Although the records represent all wild animals admitted to the rehabilitation center over a two-year period, these were the first two years in which a more detailed digital record was kept, and errors and inconsistencies occurred within the center. These ranged from misspellings to misidentified species to failure to accurately report causes of admission and outcome, which represents a potential bias. Finally, we define rehabilitation success as the release of the wild animals into nature. Although true success can be defined as normal function and survival after the moment of release, unfortunately this information is not available. Lack of follow-up is an inherent shortcoming of the current wildlife rehabilitation process. 

## 5. Conclusions

Overall, it is paramount to emphasize the great value of maintaining, improving, and studying the databases of wildlife rehabilitation centers, as they can provide useful information that can be used to enhance the allocation of economic resources, treatment methods, disease surveillance, public education and regulatory decision-making; upgrade the effectiveness of the rehabilitation process; and lead to a better understanding of threats to wildlife and subsequent implementation of conservation actions. The objective is to maximize the success of rehabilitation, as well as the survival and reproductive success of released individuals (particularly in the case of species of conservation interest), improving animal welfare and educating the public on ecological issues, which will ultimately contribute to the conservation of species. In addition, educating the public about the causes of admission of wild animals into centers and their natural behaviors could help reduce the number of animals admitted.

## Figures and Tables

**Figure 1 animals-14-00051-f001:**
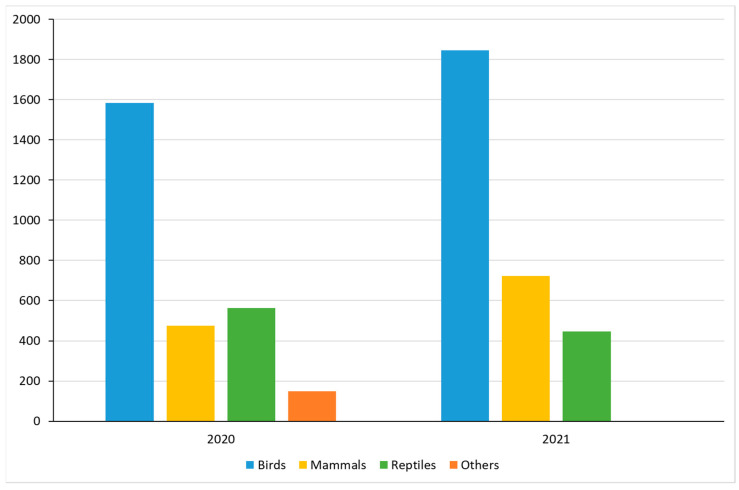
Number of rescue records per year for each class. χ^2^ test with Bonferroni correction for multiplicity.

**Figure 2 animals-14-00051-f002:**
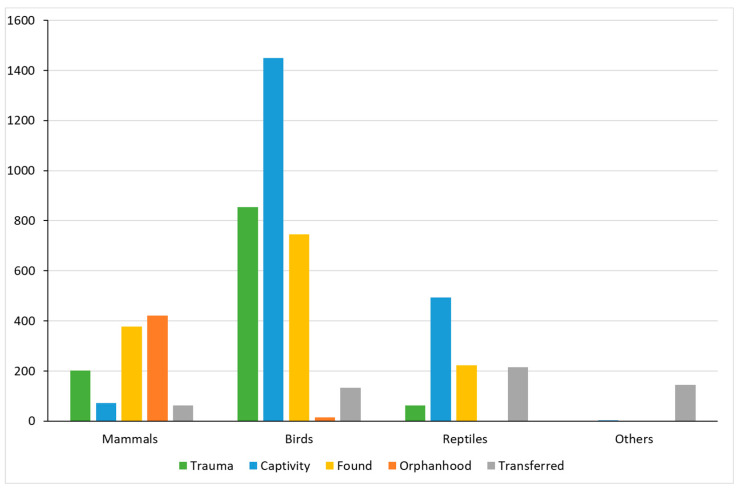
Cause of admission within each class. χ^2^ test with Bonferroni correction for multiplicity. All percentages are presented by class.

**Figure 3 animals-14-00051-f003:**
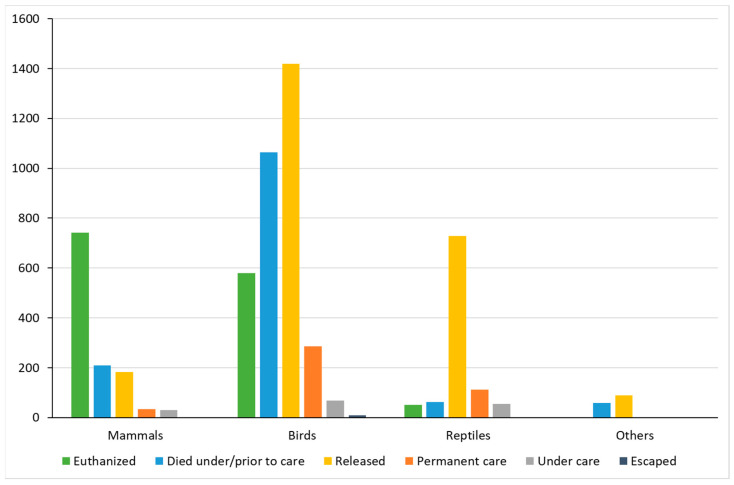
Outcomes within each animal class. χ^2^ test with Bonferroni correction for multiplicity. Values not shown correspond to 0.

**Figure 4 animals-14-00051-f004:**
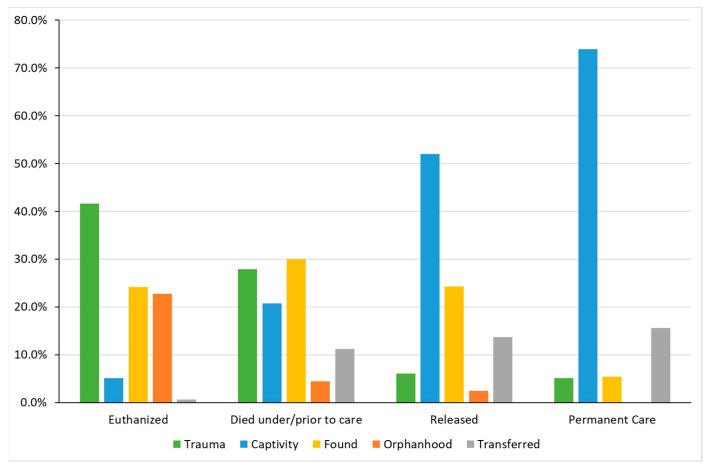
Number of rescues per cause of admission within each outcome. χ^2^ test with Bonferroni correction for multiplicity. All percentages are presented by outcome.

**Table 1 animals-14-00051-t001:** Number of rescues, orders, and species reported for each class.

Class	Number of Rescues	% of Rescues	Number of Orders	Number of Species
Birds	3430	59.3	19	187
Mammals	1196	20.7	10	41
Reptiles	1009	17.4	4	60
Others	150	2.6	3	27
Total	5785	100	36	315

**Table 2 animals-14-00051-t002:** Distribution of rescue animals per order and filo/class.

Filo/Class	Order	Number of Rescues	% of Rescues
Mammals	Didelphimorphia	578	10.0
	Pilosa	47	0.8
	Cingulata	11	0.2
	Primates	83	1.4
	Rodentia	229	4.0
	Lagomorpha	1	0.0
	Chiroptera	40	0.7
	Carnivora	179	3.1
	Perissodactyla	1	0.0
	Artiodactyla	27	0.5
Birds	Anseriformes	80	1.4
	Galliformes	30	0.5
	Pelecaniformes	8	0.1
	Ciconiiformes	40	0.7
	Falconiformes	79	1.4
	Gruiformes	15	0.3
	Charadriiformes	8	0.1
	Columbiformes	387	6.7
	Psittaciformes	1181	20.4
	Cuculiformes	16	0.3
	Strigiformes	230	4.0
	Caprimulgiformes	17	0.3
	Apodiformes	68	1.2
	Trogoniformes	1	0.0
	Coraciiformes	46	0.8
	Piciformes	112	1.9
	Passeriformes	1083	18.7
	Struthioniformes	2	0.0
	Casuariiformes	3	0.1
Reptiles	Crocodylia	14	0.2
	Testudines	691	11.9
	Suborder Lacertilia	166	2.9
	Suborder Serpentes	138	2.4
Amphibious	Anura	99	1.7
Arthropods	Lepidoptera	46	0.8
	Decapoda	1	0.0
Unknown	Unknown	28	0.5
Total		5785	100

**Table 3 animals-14-00051-t003:** Causes of admission of the rescue animals.

Cause of Admission	Number of Rescues	% of Rescues
Captivity	2020	34.9
Found	1347	23.3
Trauma	1118	19.3
Transferred	556	9.6
Orphanhood	436	7.5
Unknown	112	1.9
Disease	188	3.2
Environmental pollutants	8	0.1
Total	5785	100

**Table 4 animals-14-00051-t004:** Clinical classification of the rescue animals.

Clinical Classification	Number of Rescues	% of Rescues
Basic care	937	16.2
Clinically healthy	983	17.0
Physical injury	606	10.5
Nonspecific disease	108	1.9
Clipped feathers	46	0.8
Neurological disease	35	0.6
Nutritional disease	88	1.5
Ocular disease	22	0.4
Unknown	76	1.3
Infectious disease	16	0.3
Gastrointestinal disease	11	0.2
Dermatological disease	64	1.1
Tumors	10	0.2
Respiratory disease	3	0.1
Petrochemical exposure	7	0.1
Urogenital disease	1	0.0
Intoxication	2	0.0
Total	3015	52.1
Missing data	2770	47.9
Total + missing data	5785	100

**Table 5 animals-14-00051-t005:** Released vs. non-released rescue animals per animal class, in 2020 and 2021. Χ^2^ test with Bonferroni correction for multiplicity. All percentages showed are global.

	Released		Non-Released	
Class	N	%	N	%
Birds	1420	58.7	2006	59.7
Reptiles	728	30.1	281	8.4
Mammals	182	7.5	1014	30.2
Others	89	3.7	61	1.7
Total	2419	100	3362	100

**Table 6 animals-14-00051-t006:** Released vs. non-released rescue animals per cause of admission in 2020 and 2021. χ^2^ test with Bonferroni correction for multiplicity. All percentages are presented by causes of admission.

	Released		Non-Released	
Cause of Admission	N	%	N	%
Captivity	1234	61.2	783	38.8
Found	577	42.8	770	57.2
Transferred	324	58.3	232	41.7
Trauma	144	12.9	974	87.1
Total	2419	41.9	3297	58.1

**Table 7 animals-14-00051-t007:** Released vs. non-released rescue animals per clinical classification in 2021. χ^2^ test with Bonferroni correction for multiplicity. All percentages are presented by clinical classification.

	Released		Non-Released	
Clinical Classification	N	%	N	%
Clinically healthy	707	71.9	276	28.1
Basic care	241	25.7	696	74.3
Physical injury	66	10.9	540	89.1
Nonspecific disease	20	18.5	88	81.5
Total	1143	38.9	1796	61.1

**Table 8 animals-14-00051-t008:** Released vs. non-released rescue animals per age group in 2021. χ^2^ test with Bonferroni correction for multiplicity. All percentages are presented by age group.

	Released		Non-Released	
Age Group	N	%	N	%
Adult	239	34.8	448	65.2
Juvenile	106	41.4	150	58.6
Infant	82	15	465	85
Fledgling	72	37.9	118	62.1
Nestling	62	20.8	236	79.2
VSTotal	561	28.1	1432	71.9

**Table 9 animals-14-00051-t009:** Mortality vs. survival of the rescue animals per animal class in 2020 and 2021. χ^2^ test with Bonferroni correction for multiplicity. All percentages showed are global.

	Mortality		Survival	
Class	N	%	N	%
Birds	1644	59.4	1782	59.1
Mammals	950	34.3	246	8.2
Reptiles	113	4.1	896	29.7
Others	59	2.1	91	3.1
Total	2766	100	3015	100

**Table 10 animals-14-00051-t010:** Mortality vs. survival of the rescue animals per cause of admission in 2020 and 2021. χ^2^ test with Bonferroni correction for multiplicity. All percentages are presented by cause of admission.

	Mortality		Survival	
Causes of Admission	N	%	N	%
Trauma	938	83.9	180	16.1
Found	733	54.4	614	45.6
Orphanhood	365	83.7	71	16.3
Captivity	349	17.3	1668	82.7
Total	2704	47.7	2966	52.3

**Table 11 animals-14-00051-t011:** Mortality vs. survival of the rescue animals per clinical classification in 2021. χ^2^ test with Bonferroni correction for multiplicity. All percentages are presented by clinical classification.

	Mortality		Survival	
Clinical Classification	N	%	N	%
Basic care	647	69.1	290	30.9
Physical injury	507	83.7	99	16.3
Clinically healthy	91	9.3	892	90.7
Nonspecific disease	84	77.8	24	22.2
Total	1489	50.7	1450	49.3

**Table 12 animals-14-00051-t012:** Mortality vs survival of the rescue animals per age group, in 2021. χ^2^ test with Bonferroni correction for multiplicity. All percentages are presented by age group.

	Mortality		Survival	
Age Group	N	%	N	%
Infant	430	78.6	117	21.4
Adult	361	52.5	326	47.5
Nestling	232	77.9	66	22.1
Juvenile	130	50.8	126	49.2
Fledgling	111	58.4	79	41.6
Total	1278	64.1	715	35.9

**Table 13 animals-14-00051-t013:** Predictive parameters of mortality.

Parameter	expB	95% CI	*p*-Value
Clinically healthy	110.444	14.357–849.624	<0.001
Basic care	19.037	2.493–145.362	0.005
Reptile	5.421	2.856–10.290	<0.001
Juvenile	1.765	1.087–2.866	0.022
Days in the hospital	1.033	1.028–1.038	<0.001
Infant	0.352	0.175–0.709	0.003
Nestling	0.293	0.158–0.544	<0.001

expB—likelihood ratio, CI—confidence interval, *p*-value—probability of significance.

## Data Availability

The data presented in this study are available on request from the corresponding author. The data is not publicly available due to belonging to RWRC.

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
