# Peer review of "A Retrospective Analysis of Admission Trends and Outcomes in a Wildlife Rescue and Rehabilitation Center in Costa Rica"

_animals, 2023, doi:10.3390/ani14010051_

Round 1

Reviewer 1 Report

Comments and Suggestions for Authors

Manuscript ID: animals-2670612

Title: A two-year retrospective study on casuistry patterns of a wildlife rescue and rehabilitation center in Costa Rica

Review

The manuscript deals with wildlife rescues in the Rescate Wildlife Rescue Center in Costa Rica, analyzing causes of admission, factors of rehabilitation success, and causes of mortality of various taxonomic groups of animals. Such publications from the South America are not extensive, therefore paper would be of interest to Animals. However, in the current form submission is premature, it contains calculation errors and is too long, repeating information in textual and numerical form.

General remark:

Please stick to the Template presented in the journal: citation of references in the text; decimals separated NOT by comma, format of references, etc.

Title

two year is not a long period, therefore I propose to exclude part of the Title, namely „A two-year retrospective study on“.

Also please exclude the word „casuistry“, as it is not the term you expected:

Casuistry is a method of case reasoning especially useful in treating cases that involve moral dilemmas. It is also a branch of applied ethics.

Materials and Methods

There is no study design as such in the manuscript, as the study was retrospective. This part can be omitted/shortened. What you really do – is to enter data into Excel and analyze it.

Analysis: you use percent in most of the analysis, that is, you use proportions. Please add confidence intervals everywhere, and please use G-test to say, if these proportions are different.

As for chi-square, please explain, how you use it, as now usage is not understandable (see also comments below). E.g. – do you compare expected frequencies with the observed ones? How did you deal with different sample sizes?

Text in Lines 155–158 sounds not clear.

Results

Line 160: there is NO trend across two years.

Line 161-163: Please do not repeat the numbers of the sample size.

Figure 1: it is not clear, what differences you analyze.

Table 1,2 and others: give confidence intervals, or at least when you use data from the Tables in the text.

Figure 2: it is not clear, what differences you analyze.

Figure 3: not needed, you give all data in the text. Please add CI and G-test if intergroup differences are significant.

Figure 4: it is not clear, what differences you analyze. Output is doubtful – one asterisk in mammals, so aren’t the difference between “released” and “permanent care” significant? Why three asterisks for birds? It looks like all frequencies in birds differ from each other.

Figure 5: post-hoc Sidak not documented before. Check the spelling of this test in the initial reference.

Figure 2: it is not clear, what differences you analyze. Maybe use letters, then it will be clear about differences between groups, or, add CI to the graph. No need to show every percentage on the graph, and please do not repeat these in the text.

Table 5: based on the G-test, all four groups differ, now asterisk is used in the wrong way.

Tables 6, 7, 8, 11, 12, 13: wrong, numbers do not correspond to percentages, their total is not 100%, and the sum of numbers is also wrong.

Figures 7 and 8: not needed, just give numbers in the text.

References

At least some comparison with other databases on wildlife rehabilitation must be done. I can recommend some sources:

https://doi.org/10.1371/journal.pone.0257675

https://doi.org/10.1371/journal.pone.0181331

https://doi.org/10.1371/journal.pone.0257209

https://doi.org/10.1016/j.jnc.2020.125897

https://doi.org/10.1016/j.jnc.2023.126372

Further review of the manuscript is not appropriate, as errors in calculations are reflected in the text, and, possibly in Discussion/Conclusions. Text will be changed after recalculations.

Comments on the Quality of English Language

-

Reviewer 2 Report

Comments and Suggestions for Authors

General comments

This is potentially a very nice and useful paper to add to the literature becoming available on wildlife rehabilitation around the world. 

I have suggested some revisions are needed, but most of this is reducing your results section and focusing on what is important.

I don’t think it’s a problem having missing data from 2020, I really understand how difficult collecting this data is in these types of setting. You must however make it clear throughout when you are considering 2020+2021 data and when just 2021 data.

It feels like the early data analyses in the results are well written up and interpreted in the discussion, but the later data analyses in the results all seem a bit too much for you to properly discuss. It’s like you have ‘death by data’. Do you really need all the data in your results? Maybe just having the important things and discussing them well would be better?

There’s a lot of discussion about the need for databases, but not much about why. Are these just useful for conservation? You briefly mention ‘triage’ but don’t really explain this. Where does welfare come into all this, is welfare not the main driver for wildlife rehabilitation In most instances (alongside education)? 

Line 8              There are actually quite a few studies internationally that you haven’t referred to and might have been useful for comparison to your own work. I’ll list some other references at the end.

Line 9              Not sure about ‘menaces’ or quite what you are trying to stress. ‘Impacts’ maybe, ‘anthropogenic impacts’, 

Line 10 ‘           Casuistry patterns’ is needed or an alternative

Line 15             ‘Datebases’ - other centres have them too

Line 28            What are ‘others’ amphbians?

Line 43            the definition is ‘The treatment and temporary care of injured, diseased, and dis- placed indigenous animals, and the subsequent release of healthy animals to appropriate habitats in the wild.’ You’ve missed a bit at the beginning’

Line 45            ‘free-living’ might be better

Line 48            Why must euthanasia be considered at all stages? Maybe welfare?

Line 52            Remove ‘Therefore’ as the two statements are not related. The sentence about wildlife rehab. (line 50) might be better as the last sentence of this paragraph.

Line 60            The number ‘rehabilitated’ or rehabilitated to the point of release? When does ‘success’ happen?

Line 63-17       This is really good, all about the importance of triage and its impact on both welfare and resources. Pick up these themes in the discussion, reflecting on your own data.

Line 75            Maybe more important to minimise stress and release as soon as possible, rather than to follow protocols. Protocols might not help.

                        I’m not sure that Protocols for wildlife rehabilitation have generally been standardised. The reference is more about the need to maintain natural behaviours, whilst ‘Protocols’ suggests details for the whole process. Maybe say it is important to establish standard protocols within wildlife centres that consider things such as maintaining natural behaviours - this fits well with what you then say at Line 81-94 which is excellent.

Line 79            Not sure I agree that ‘they will never be in as good shape as if they had remained wild’ or that is what JK says in the reference. Surely the whole point of admitting them is that they would have died if not admitted and the aim is to get them back to a state where they have the same chance of survival as any other animal of that species?

Line 96/97       It’s not that treatment should be kept to a minimum - treatment should be as good as possible to ensure release. Time in captivity should be kept to a minimum, or more animals should be released as soon as they are fit.

Line 97            ‘Carers’ would probably be a better English work, or ‘keepers’ than ‘caretakers’

Line 98            ‘proper preparation for release’ - assume this means ‘rehabilitation’?

Line 116/117   Would be interesting to know ‘who’ carries out the assessment. Is this vet, trained staff members, students? You make it clear for the behavioural assessment.

Line 119          the word ‘held’ might be better as ‘triaged’

Line 123          I assume it is illegal generally to release these species back to the wild rather than it being this specific centres isn’t allowed to release?

Line 131          I’m not sure I understand what ‘and the time that was a mascot’ means?

Line 131          Do you need to say ‘The data provided by the RWRC has been corrected.’? Did you actually correct data or did you just exclude some things? If just excluding then the sentence about correcting data isn’t needed - it does make it sound like you amended the data in some way.

Line 147          That’s a lot of missing data. As well as not being in the graphs, how can you interpret these things without that data?

Line 159          General comments on the results. It starts off really well with some explanation of what you are looking at, but then ends with just tables that are hard to put into any context. You also don’t make it clear when you have two years of data and when you just have one. It might be best to look at all 2yr categories first and then do the extra 2021 categories and analyses at the end of that. I actually wonder if you want to include all this data analysis and if it’s all really relevant to your discussion? This section needs some work.

Line 160          Were there any significant differences between years? Probably not, but good to check and say.

Line 182          You don’t say in you methods (and maybe you should), but how were these reasons determined? Was it just based on what the finder of the animal said, was that always easy to determine, what happened if animals fit more than one reason e.g. they were an injured orphan?

Line 194          On my version of the paper I’m really struggling to see where the asterisk are indicating statistical significance. Can you move them closes to the relevant column and make them bigger maybe? It might also be good to mention the main significant differences in the text.

Line 197          Saying half your data is missing really makes the data useless. However, you have all the data from 2021 so say that instead. I’d not include 2020 at all in this section and would make it clear this section was just 2021 e.g. ‘Table 4 shows the clinical classification of the rescued animals in 2021’

Line 217          The asterisk are good here  Again I might briefly describe the significant differences in the main text.

Line 219          Is the average number of days calculated from the ‘final date’. If so these figures must just be 2021 and you need to say that. Otherwise the reader presumes these figures are for both years and that’s then not correct.

Line 226          Extra space in this line needs correcting

Line 255          Maybe make it clear this is just 2021 animals

Line 261          Maybe make it clear this is just 2021 animals

Line 268          It seems off to compare age to outcome, when age alone hasn’t been covered. You only have age data for 2021 as well don’t you, if so you need to say so. 

Line 317          I don’t think you need the percentages in the discussion as they are in the results section

Line 326          The discussion is really interesting and your passion for the subject comes through. As I mentioned at the start, I might just remove the results that are not key to what you want to discuss and make more of your main finding, comparing them to other studies and telling us all the good interesting things about wildlife in Costa Rica.

Line 362          There are lots of other studies relevant here, not just the one in the UK and the one in Australia referenced. 

Line 379          Is this just ‘in 2021’?

Line 393          Might be interesting to know what sorts of anthropogenic trauma occur in Cost Rica even if they are all lumped together as ‘trauma’

Line 397          It would be interesting to know what the diseases were, perhaps from a conservation perspective. However if the animals are euthanased anyway is that necessary and a good use of resources from a welfare perspective? Could more techniques be taught to staff and students like post mortem examinations?

Line 443          This section is great as a review of the other available literature, but it needs to be related to what you found in your study. It feels like you haven’t interpreted all those graphs at the end of your results. 

Line 450          This is the first time you’ve mentioned the work triage! This this needs more discussion.

Line 492          The conclusion doesn’t say much about your own findings and then gets a bit waffly.

Examples of other references to perhaps consider. These are general ones, there are lots more if you start considering specific order or species.

Cope HR, McArthur C, Dickman CR, Newsome TM, Gray R and Herbert CA 2022 A systematic review of factors affecting wildlife survival during rehabilitation and release. PLoS ONE 17(3): e0265514. https://doi.org/10.1371/ journal.pone.0265514

Hanson M, Hollingshead N, Schuler K, Siemer WF, Martin P and Bunting EM 2021 Species, causes, and outcomes of wildlife rehabilitation in New York State. PLoS ONE 16(9): e0257675. https://doi.org/10.1371/journal.pone.0257675

Kwok ABC, Haering R, Travers SK and Stathis P 2021 Trends in wildlife rehabilitation rescues and animal fate across a six-year period in New South Wales, Australia. PLOS ONE 16(9): e0257209.

https://doi.org/10.1371/journal.pone.0257209

Montesdeoca N, Calabuig P, Corbera JA, Cooper JE and Orós J 2017a Causes of morbidity and mortality, and rehabilitation outcomes of birds in Gran Canaria Island, Spain. Bird Study 64(4): 523-534. https://doi.org/10.1080/00063657.2017.1411464

Romero F, Espinoza A, Sallaberry-Pincheira N and Napolitano C 2019 A five-year retrospective study on patterns of casuistry and insights on the current status of wildlife rescue and rehabilitation centers in Chile. Revista Chilena de Historia Natural 92(6). https://doi.org/10.1186/s40693-019-0086-0

Schenk AN and Souza MJ 2014 Major anthropogenic causes for and outcomes of

wild animal presentation to a wildlife clinic in East Tennessee, USA, 2000–2011. PloS One 9: e93517. https://doi.org/10.1371/journal.pone.0093517

Taylor-Brown A, Booth R, Gillett A, Mealy E, Ogbourne SM, Polkinghorne A and Conroy GC 2019 The impact of human activities on Australian wildlife. PLoS ONE 14(1): e0206958. https://doi.org/10.1371/journal.pone.0206958

Wimberger K and Downs C 2010 Annual intake trends of a large urban animal rehabilitation centre in South Africa: A case study. Animal Welfare 19(4): 501-513. https://doi.org/10.1017/S0962728600001974

Wimberger K, Downs CT and Boyes RS 2010 A survey of wildlife rehabilitation in South Africa: Is there a need for improved management? Animal Welfare 19: 481–499. https://doi.org/10.1017/S0962728600001962

Comments on the Quality of English Language

The English is generally very good. There are a few words I've flagged which are perhaps not correct. Getting a native English speaker to read over the final draft would be useful.

Author Response

Thank you very much for taking the time to review this manuscript. I agree with all of the comments that were made. Therefore, I have made major changes to the manuscript. 

Best regards, 

Maria Costa 

Reviewer 3 Report

Comments and Suggestions for Authors

The manuscript provides an informative analysis of the causes and outcomes of wildlife rescue by an organisation in Costa Rica. There is no replication of wildlife rescue centres in the dataset, but the sample size is large and covers the geographic area of Costa Rica. Even so the authors refer to ‘transfer from another center’ in Table 3 and should enlighten the reader about such centres (wildlife rescue, law enforcement, customs and border security?) to determine if there was opportunity for replication and/or whether biases could arise from treatment in prior centres. Otherwise, the methodology is sound with a basic but appropriate statistical analysis. The results exhaust the dataset which in this case is not excessive. However, the presentation can be improved by being more succinct and the colours in the bar chart figures consistent solid ones. For example, it is better to be direct – This happens (Fig. X) – rather than – Figure X shows this happens. This better keeps the readers’ attention rather than labouring through unnecessary text. The discussion is enlightening and makes appropriate commentary on the study’s limitations and opportunities arising for further research. This is ably captured in the conclusions. Citations are not always consistent with the MDPI style and items 21 and 28 do not give full authorship which is sloppy.

There are several minor improvements that can be made which I list below. If the authors are diligent in correcting the manuscript, then it should not need further external review.

Line 8: ‘careful examination’ is vague perhaps ‘comprehensive analysis’

Line 10: objective criteria for what?

Line 10: Review the definition of ‘casuistry’ in say the Oxford English dictionary. It is often used in politics to suggest deception but also has a more philosophical meaning. Neither are appropriate here so perhaps ‘causation’?

Line 46: It? Do you mean ‘survival’?

Line 47: ‘to nature’? Do you mean ‘to living in the wild’?

Line 65: it is that the animal will be

Line 67-68: More research should be dedicated to…and thus reduce the potential for

Line 69: ,and to improve

Line 79: ‘never be in as good shape’? Not necessarily true if, for example, they have a better plane of nutrition in a brief captive period which gets them over a developmental hump whereas in the wild they may have starved or have delayed development.

Line 83: that have performed

Line 104: this study are (the article simply reports it)

Line 116: arrival they are

Line 123: to release

Line 131: that institution was a mascot. Mascot? Do you mean supporter, or donor or fund-raiser? Supporter is the best generalisation.

Line 165: to bird and mammal rescues

Line 186: Not clear what ‘ambiental’ in table means?

Line 201:  Infectious disease

Line 228: Not clear what the (>5%) implies given values like (o.6%)?

Line 333: with parrots

Line 369: as confirmed

Line 399: funding

Line 410: offspring than just

Line 425: since triage

Line 441: on humans

Line 471: bonds to conspecifics

Comments on the Quality of English Language

See above

Author Response

Thank you very much for taking the time to review this manuscript.  I agree with all your comments. Therefore, I have corrected all the errors that were pointed out. 

Round 2

Reviewer 1 Report

Comments and Suggestions for Authors

Please find round 2 comments attached

Comments on the Quality of English Language

I am not responsible for language quality of the manuscript, as for reading it seems more or less ok.

Author Response

Thank you so much for taking the time to review this manuscript. I hope you find the changes I made suitable. 

Best regards, 

Maria Costa

Reviewer 2 Report

Comments and Suggestions for Authors

Very much improved and now looks great and reads very well. Very well done.

Author Response

Thank you so much for reviewing this manuscript and for your insightful comments. 

Best regards, 

Maria Costa 

Round 3

Reviewer 1 Report

Comments and Suggestions for Authors

Thank you. Now I understanf, that in the tables 5 to 12 you compare every line of two cilumns. Just think, if captions say this clearly. 

Comments on the Quality of English Language

Language is understandable, maybe MDPI wull polich it while doing la.out